# Revisiting the wintertime emergent constraint of the Southern Hemispheric midlatitude jet response to global warming

Philipp Breul[1], Paulo Ceppi[1,2], and Theodore G. Shepherd[3]

[1]Department of Physics, Imperial College London, London, United Kingdom
[2]Grantham Institute, Imperial College London, London, United Kingdom
[3]Department of Meteorology, University of Reading, Reading, United Kingdom

**Correspondence:** Philipp Breul (pyb18@ic.ac.uk)

**Abstract.** Most climate models show a poleward shift of the southern hemispheric zonal-mean jet in response to climate change, but the inter-model spread is large. In an attempt to constrain future jet responses, past studies have identified an emergent constraint between the climatological jet latitude and the future jet shift in austral winter. However, we show that the emergent constraint only arises in the zonal mean, and not in separate halves of the hemisphere, which questions the physicality of the emergent constraint. We further find that the zonal-mean jet latitude does not represent the latitude of a zonally coherent structure, due to the presence of a double jet structure in the Pacific region during this season. The zonal asymmetry causes the previously noted large spread in the zonal-mean climatology but not in the response, which underlies the emergent constraint. We therefore argue that the emergent constraint on the zonal-mean jet cannot narrow down the spread in future wind responses, and propose that emergent constraints on the jet response in austral winter should be based on regional rather than zonal-mean circulation features.

## 1   Introduction

The southern hemispheric midlatitude jet is predicted to shift poleward in response to greenhouse gas forcing. However, the magnitude of this shift differs among global climate models (GCMs) (Barnes and Polvani, 2013; Curtis et al., 2020). This in turn increases the uncertainty of projected climate change impacts on the mid-latitude region (Shepherd, 2014). It is therefore necessary to constrain the range of future jet responses.

One way of narrowing down the range of model responses is by identifying emergent constraints (EC), which are across-model relationships between a climatological variable $X$ and the response in the variable of interest $Y$ (Hall et al., 2019). If such a constraint is found, it can be used by calculating $X$ from real world data to predict the response in $Y$. A commonly cited example involves the correlation between climatological jet latitude and future jet shift across CMIP models in southern hemispheric winter, first identified by Kidston and Gerber (2010). As a physical explanation, the authors proposed Fluctuation-Dissipation Theory, which in a simplified form links future jet shift to an annular mode timescale (e.g. Ring and Plumb, 2008; Breul et al., 2022). However, Simpson and Polvani (2016) cast doubt on these findings, since they found the inter-model spread in annular mode timescale to have opposite seasonality to the correlation strength between jet latitude and shift. They further

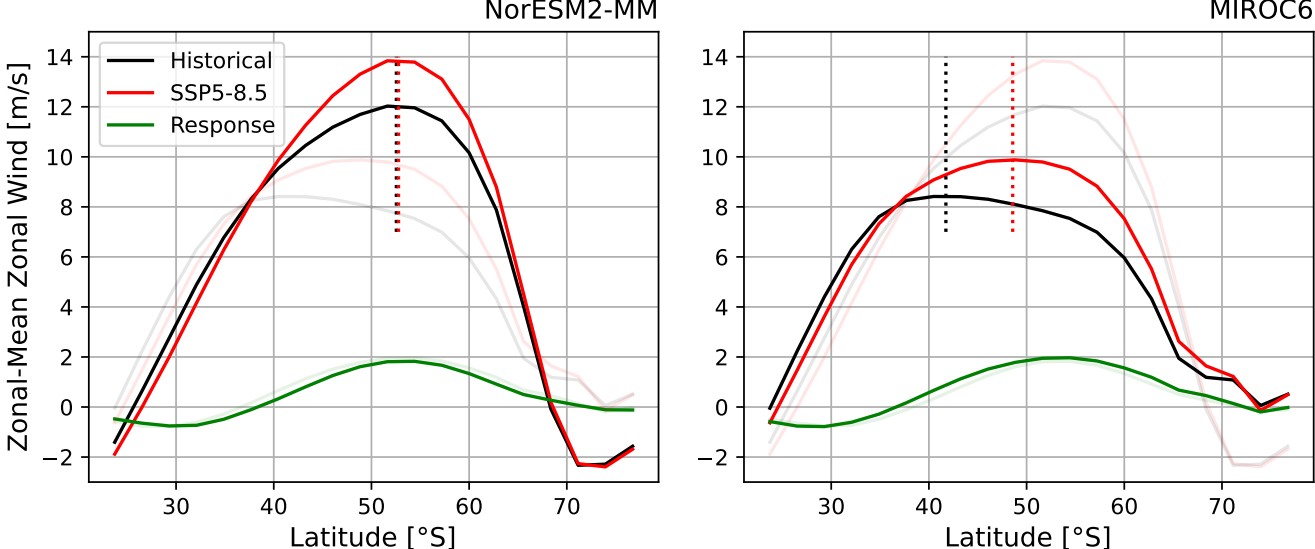

**Figure 1.** JJA zonal-mean zonal wind climatology of Historical and SSP5-8.5 scenarios and their difference for two selected CMIP6 models: **(a)** NorESM2-MM and **(b)** MIROC6. Dotted vertical lines denote the jet latitude. For better comparison both (a) and (b) also show the curves of the respective other model as shaded lines.

25  found the relationship between jet latitude and future jet shift to hold in winter only but not summer.

More recently, Curtis et al. (2020) and Simpson et al. (2021) confirmed the existence of the same wintertime constraint in CMIP6. Simpson and Polvani (2016) proposed a possible explanation for the EC by observing that the zonal wind response does not track the climatological jet latitude, as one would expect if the response were always a shift of the jet, but is approxi-
30  mately the same and independent of initial jet latitude. They speculated that this effect gives rise to the EC, since the response projects more or less strongly onto a jet shift, depending on its climatology. We demonstrate this effect in Fig. 1 for two CMIP6 models with almost identical zonal-mean zonal wind responses but different climatologies, which leads to a large difference in future jet shift. However, the reason for this "anchoring" of the response remains unclear.

35  An understanding of the physical basis of an EC is important for having confidence in its ability to constrain future responses (Hall et al., 2019). Here we propose that both the EC and the anchored zonal wind response can be explained by a geometric argument, based on the zonal-mean jet latitude not reflecting the position of a coherent structure in wintertime because of zonal asymmetries associated with a double jet structure in the Pacific region. This questions the physical basis (and therefore the usefulness) of the zonal-mean jet latitude as a circulation metric in wintertime, and consequently of the EC.

## 2 Data and Methods

We use the austral wintertime (June–July–August, JJA) zonal wind at 850 hPa, regridded to a common T42 grid, from 39 models participating in the historical and SSP5–8.5 experiments of the Coupled Model Intercomparison Project Phase 6 (CMIP6), detailed in Table A1.

We use the periods 1950–2014 for the historical experiment and 2076–2100 for the SSP5-8.5 experiment, and the response is defined as the climatological difference between the two. The results presented are quantitatively the same when other time periods are chosen. Unless otherwise specified, we always consider the wintertime seasonal average. The data is restricted to the latitude range $22°$S – $78°$S. The jet latitude is defined as the maximum of a parabola fitted to the maximum zonal-mean zonal wind grid point and its two neighbours, as was done by Kidston and Gerber (2010). The jet shift is then the difference in climatological jet latitude between the historical and SSP5-8.5 experiments.

When determining whether multiple jets occur at the same time, we identify local maxima in the daily zonal-mean zonal wind data that are spaced at least 5 grid points in latitude apart (approximately $14°$) and have a strength of at least 4 m/s. To filter out eddy contributions, we use a Butterworth low-pass filter with a cutoff frequency of $(8 \, \text{days})^{-1}$ on the zonally resolved data. This is the only part of the analysis where we use daily data rather than seasonal averages. Owing to limited data availability, this analysis was restricted to 29 models, detailed in Table A1.

## 3 Results & Discussion

### 3.1 Emergent Constraint

First, we reproduce the wintertime EC that was found by Simpson and Polvani (2016) in CMIP5, and Simpson et al. (2021) in CMIP6, between the climatological jet latitude and the future jet shift in Fig. 2a. While we find a Pearson correlation coefficient of $r = -0.76$, the data includes outliers. Nevertheless, measuring the correlation strength with the Spearman rank correlation, which is less sensitive to outliers, still gives a high value of $\rho = -0.66$.

However, this relationship was obtained in the zonal mean. Fig. 3a shows the CMIP6 model average of longitudinally resolved zonal wind, which shows a clear asymmetry in the hemisphere, with a double jet structure in the Pacific region. Therefore we repeat the analysis in Fig. 2b and c for the Indo-Atlantic region ($300°$– $120°$) and the Pacific region ($120°$– $300°$) separately. We note that the exact hemispheric division is not overly important; the values presented here were chosen since they cover the full longitudinal extent of the double jet structure in the climatological and model mean, while at the same time dividing the hemisphere into two equal parts. The Indo-Atlantic region clearly does not exhibit an EC, and features relatively little variation in both mean jet latitude and jet shift. While the Pearson correlation coefficient is high in the Pacific region, this effect seems to depend on a few large outliers (note the larger $y$-axis range in Fig. 2c). The Spearman rank correlation is indeed significantly

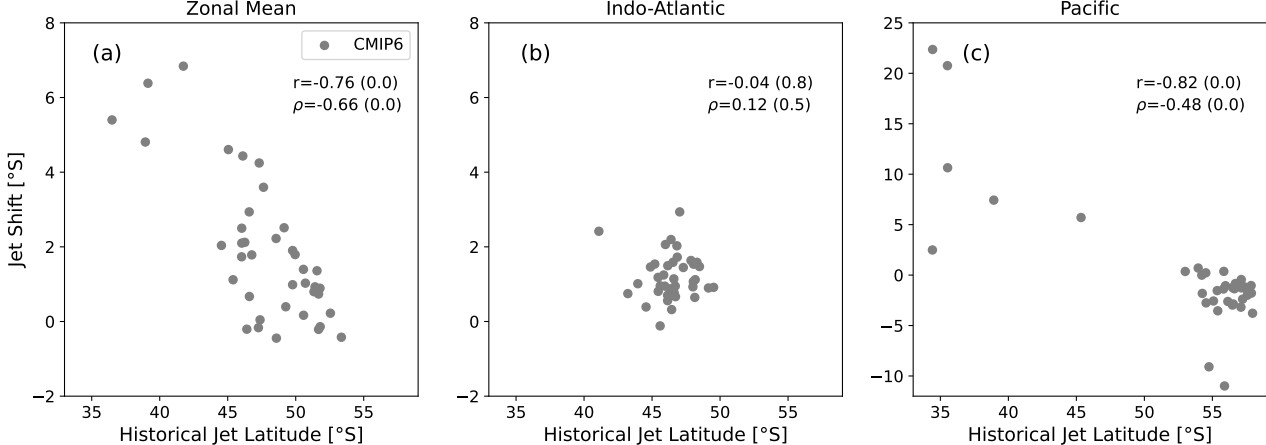

**Figure 2.** JJA future jet shift plotted against climatological jet latitude for the CMIP6 models. The Pearson correlation coefficient $r$ and the Spearman rank correlation coefficient $\rho$ are given for the CMIP6 values together with their respective $p$-values. The zonal mean was taken over **(a)** all longitudes, **(b)** $300° - 120°$ and **(c)** $120° - 300°$. Note the larger $y$-axis range in (c).

weaker, and when excluding the outliers (historical jet latitude equatorward of $50°$S or jet shift greater in magnitude than -5°S) we only find small correlation coefficients with high $p$-values. Note that we do not find a significant correlation when repeating the analysis for each jet of the double jet structure separately. This raises the question of where the EC comes from and why it
only appears in the whole hemispheric zonal mean.

## 3.2 Climatological Jet Latitude

The origin and behaviour of the asymmetry introduced in the previous section has been investigated in several previous studies (e.g. Inatsu and Hoskins, 2004; Codron, 2007; Patterson et al., 2020). To verify that the Pacific double jet structure exists physically and is not just an artefact of time and model averaging (e.g. in case of bimodality in the latitudinal distribution of
a single jet), we analyse daily data of the Pacific and Indo-Atlantic sectors. We identify the number and location of jets (as described in Section 2) in each sector for all historical days and models. For the Pacific sector we find a single jet 38.7% of the time and a double jet 57.3% of the time, while the residual 4% showed three peaks or more. We show the latitude probability density for single- and double-jet situations in the Pacific in Fig. 3b and c respectively. This shows that a double jet structure physically exists most of the time in the Pacific region, different from the Indo-Atlantic sector which shows a single jet 79.2%
of the time and a double jet only 20.6% of the time.

The observed zonal asymmetry means the zonal-mean jet latitude does not reflect the location of a zonally coherent structure. We sketch this effect in Fig. 4a, where we represent the observed jets using Gaussians. For demonstration purposes the two jets of the double jet structure are positioned slightly further apart than what is typically found in CMIP models, but otherwise
the structures shown are realistic and fit in the range of model behaviour; see Figs. 5 and B1. Figure 4a demonstrates how

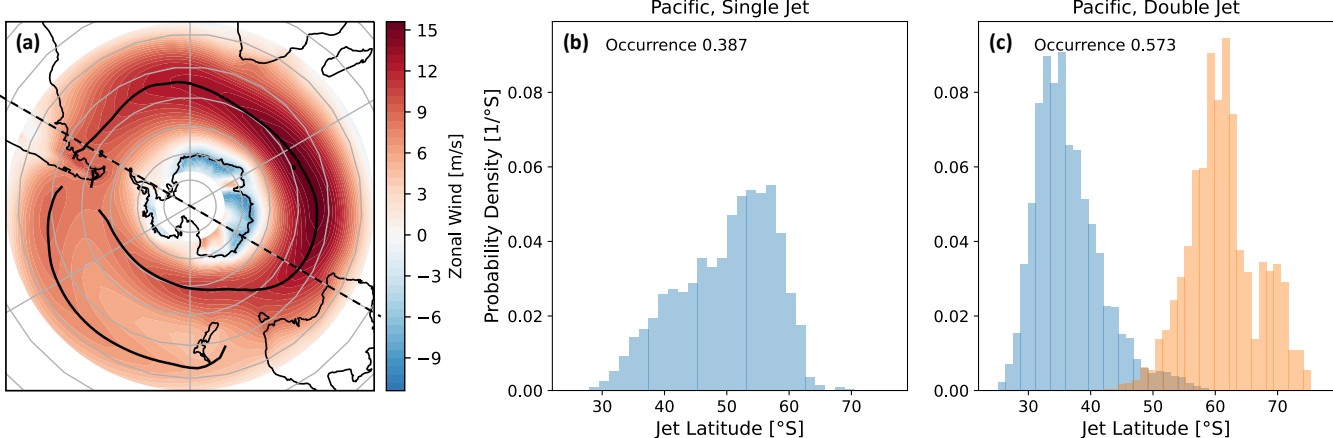

**Figure 3. (a)** 850 hPa climatological zonal wind averaged over all CMIP6 models; the solid black lines trace the jet positions. The dashed black line divides the hemisphere into the Indo-Atlantic and the Pacific sectors. **(b)** Jet latitude probability density for days with a single jet structure in the Pacific; **(c)** same as (b) for days with a double jet structure.

differences in the strength of the individual jets in the Pacific double jet structure lead to different zonal-mean jet latitudes, even though none of the jet structures have moved. To test whether this effect is present in the CMIP6 ensemble, we plot the historical zonal-mean jet latitude against the difference in strength between the two Pacific jets for the CMIP6 models in Fig. 4b, where we find a strong correlation. Meanwhile, we find only weak Spearman correlation coefficients with high
$p$-values between the zonal-mean jet latitude and either of the Pacific jet latitudes. This supports the hypothesis that differences in the Pacific double jet strength are able to account for the differences in the zonal-mean jet latitude, which could explain the larger spread in jet latitude in the zonal mean compared to the Indo-Atlantic sector. The analysis was limited to the 16 models that show two distinct peaks in the zonal-mean climatology over $120° - 300°$  for simplicity; using daily data a similar analysis could likely be performed for all models.


As an aside, we note that the spread in jet latitude over the Indo-Atlantic half of the hemisphere also contributes to the inter-model spread in zonal-mean jet latitude, as can be seen in Fig. 4c where we regressed the inter-model differences in zonal wind onto the inter-model differences in jet latitude. We further find an increase in the Indo-Atlantic mean jet latitude to be correlated with a strengthening of the mean poleward Pacific jet. However, this connection does not invalidate our point that the
zonal-mean jet latitude does not reflect the latitude of any individual jet structure in the Pacific sector. Hence, in the following discussion we will focus on the Pacific contribution for reasons of simplicity.

The results so far therefore suggest that the zonal-mean climatological jet latitude does not reflect the position of a zonally coherent structure. This implies that any analysis involving this measure of jet latitude should be interpreted with caution,
including the EC shown in Fig. 2a (especially since it is not present in the separate halves of the hemisphere).

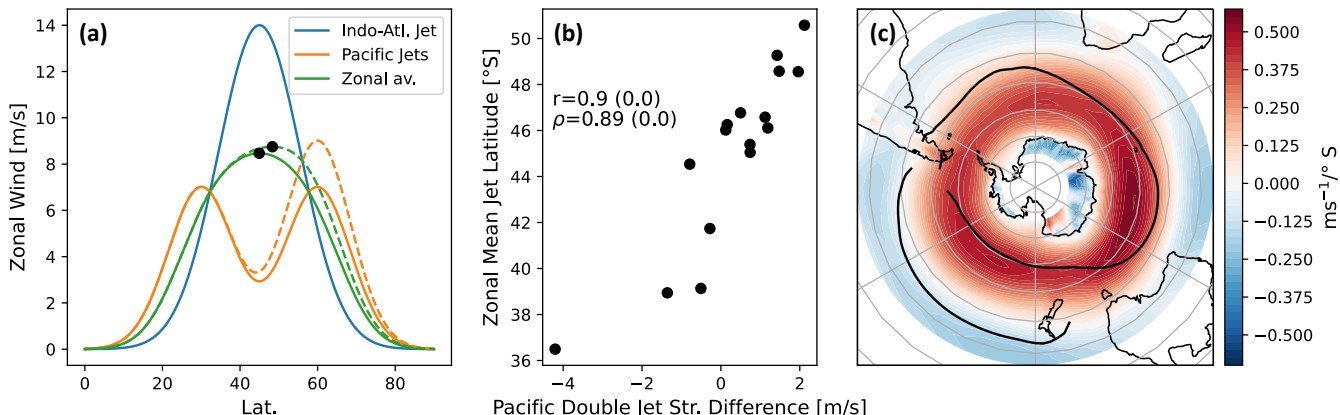

**Figure 4. (a)** Idealised sketch of the zonal-mean zonal wind structures in the Indo-Atlantic and Pacific sectors and their zonal average. The dashed lines denote a different climatology, while the black dots mark the jet peaks in the two climatologies. **(b)** The difference in strength between the Pacific double jets plotted against the zonal-mean jet latitude for the 16 CMIP6 models that show two distinct peaks in the Pacific zonal-mean climatology. **(c)** Inter-model differences in zonal wind climatology regressed onto the inter-model differences in zonal-mean jet latitude. The solid black lines trace the jet positions as in Fig. 3a.

### 3.3 Toy Model

In this section we introduce a toy model to propose a mechanism that could cause the zonal-mean EC. The basis for the model are the same structures shown in Fig. 4a, i.e. a single jet structure for the Indo-Atlantic half and a double jet structure for the Pacific half, which are then averaged together. The jets are represented by Gaussians of the form

$$g(\theta) = a \cdot e^{-\frac{(\theta - \mu)^2}{\sigma^2}}, \tag{1}$$

with $\theta$ the latitude, $\mu$ the jet latitude, $\sigma$ the width and $a$ the maximal jet strength. The Pacific double jet is represented as the sum of two Gaussians. We set the values such that they fit the CMIP6 average for both the Pacific sector (see Fig. 5) and the Indo-Atlantic sector (see Fig. B1) for the historical and SSP5-8.5 scenarios. The values can be found in Table 1. For the Pacific double jet structure, we use the subscripts 1 and 2 to refer to the equatorward and poleward jet, respectively.

To simulate differences in climatological jet latitude created by differences in double jet strength (as shown in Fig. 4b), the equatorward Pacific jet strength $a_1$ is made up of a base value with an added random variable $r_1$ equally distributed between $(-1.5, 1.5)\,\mathrm{m\,s^{-1}}$. Similarly the poleward Pacific jet strength $a_2$ is set to a base value plus a random variable $r_2$ equally distributed between $(-3, 3)\,\mathrm{m\,s^{-1}}$. The variations in the Indo-Atlantic mean state were found not to be qualitatively important and were therefore not included in the toy model for simplicity. In this manner we create 39 different realisations that emulate the range of CMIP6 model behaviour seen in Fig. 5a and b (light red lines). While this model cannot describe every aspect of the inter-model differences in jet climatology, we believe it captures the essential features for our problem, while at the same time being relatively simple. By design, the spread in toy model responses is minimal, since the same change in parameters is

|  | Indo-Atlantic | Pacific |
|---|---|---|
| Historical | $a = 13.5\,\frac{m}{s}\,, \mu = 47°, \sigma = 15.5°$ | $a_1 = 7\,\frac{m}{s} + r_1, \mu_1 = 37°, \sigma_1 = 12°$ <br> $a_2 = 8.5\,\frac{m}{s} + r_2, \mu_2 = 57°, \sigma_2 = 12°$ |
| SSP5-8.5 | $a = \mathbf{14.5}\,\frac{m}{s}\,, \mu = \mathbf{48°}, \sigma = 15.5°$ | $a_1 = 7\,\frac{m}{s} + r_1, \mu_1 = \mathbf{39°}, \sigma_1 = 12°$ <br> $a_2 = \mathbf{10}\,\frac{m}{s} + r_2, \mu_2 = 57°, \sigma_2 = 12°$ |

**Table 1.** Toy model parameters from Eq. (1) used to fit the Indo-Atlantic jet (see Fig. B1) and the double jet structure in the Pacific (see Fig. 5) for both the historical and SSP5-8.5 scenarios. The strengths of the Pacific double jet structure $a_1$ and $a_2$ are the base values, to which a random perturbation is added in each model realisation. The subscripts 1 and 2 refer to the more equatorward and more poleward Pacific jet, respectively. The parameter values that change to approximate the SSP5-8.5 response are highlighted in bold.

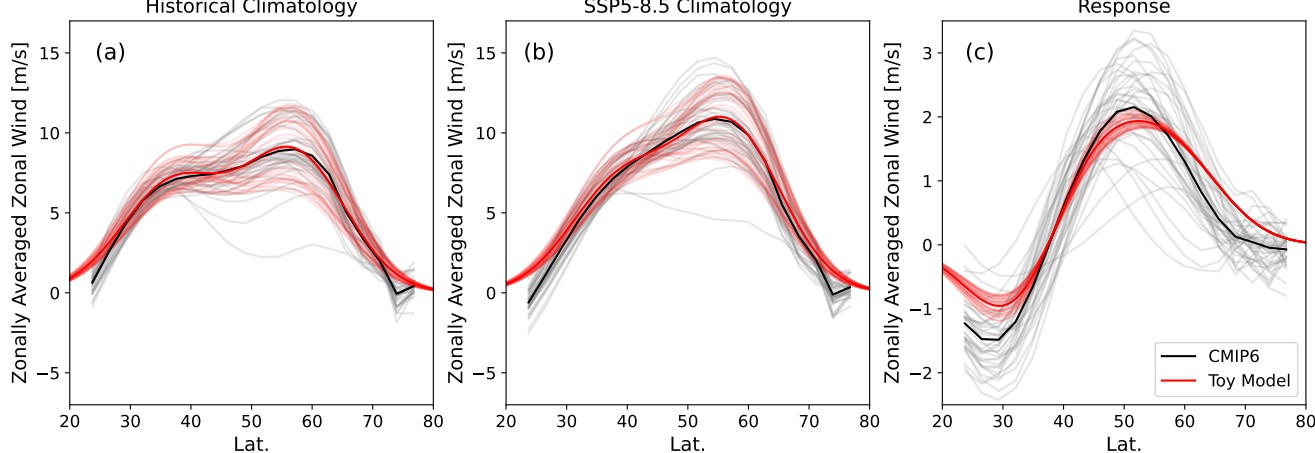

**Figure 5.** Zonal-mean zonal wind in the Pacific sector ($120° - 300°$); individual CMIP6 models in grey and their average in black. In light red are the individual realisations of the toy model (see Section 3.3) and in dark red their average. **(a)** Historical period, **(b)** the SSP5-8.5 scenario, and **(c)** the climate-change response as the difference between (b) and (a).

used for all realisations – see Fig. 5c for the Pacific sector and Fig. B2c for the zonal mean.


The parameters of the toy model are given in Table 1. We note that the change in parameters between the historical and SSP5-8.5 scenarios signifies a poleward shift and strengthening of the Indo-Atlantic jet structure, and in the Pacific basin a poleward shift of the equatorward jet and a strengthening of the poleward jet. These conclusions are qualitatively in agreement with the changes observed in the jet strength and location histograms derived from daily data (not shown). Additionally, we note that

the toy model results discussed below are not qualitatively dependent on including the response from the Indo-Atlantic sector, although including it brings the results into closer quantitative agreement with the CMIP6 ensemble.

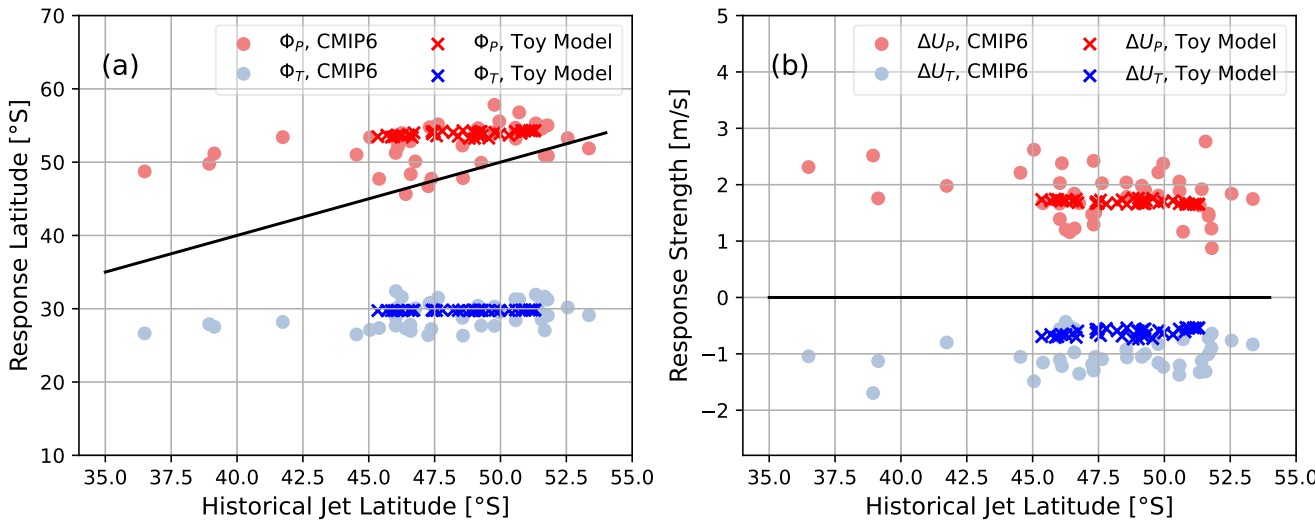

**Figure 6.** Comparison of the response structure of the CMIP6 ensemble with the toy model results. **(a)** $\Phi_P$ and $\Phi_T$, respectively the latitude of the peak and trough of the zonal wind response, against the historical jet latitude. The black line shows the identity. **(b)** $\Delta U_P$ and $\Delta U_T$, respectively the amplitude of the peak and trough of the zonal wind response, against the historical jet latitude. The black line marks zero response strength.

### 3.4 Response Structure

We now turn to testing whether the toy model can reproduce the observed inter-model relationships in the zonal-mean jet response. For this we reproduce prior analyses from Simpson and Polvani (2016) and Simpson et al. (2021) by plotting the response peak and trough locations ($\Phi_P$ and $\Phi_T$), as well as their strength ($\Delta U_P$ and $\Delta U_T$), against the historical jet latitude (Fig. 6). We observe that the response peak and trough in Fig. 6a generally do not follow the identity line (although the response peak $\Phi_P$ does to an extent), but stay fixed or "anchored". Additionally, the strength of the response is not correlated with the historical jet latitude, as shown in Fig. 6b. The toy model shows both the anchoring of $\Phi_P$ and $\Phi_T$ (Fig. 6a) as well as the asymmetry in response strength between $\Delta U_P$ and $\Delta U_T$ (Fig. 6b).

The toy model reproduces the response anchoring for two reasons. First, it simulates part of the large spread in zonal-mean jet latitude by perturbing the strength of the Pacific double jet structure, which is in line with the previous findings for CMIP6 (Section 3.2). Second, the toy model uses the same response for all realisations, derived from the model-mean climatological change in the two halves of the hemisphere. These two effects taken together lead to the toy model showing an anchored response in Fig. 6a. We note that unlike the toy model, the CMIP6 ensemble does show inter-model differences in the zonal-mean response (Fig. B2c), which is the reason for the larger spread of $\Phi_P$ and $\Phi_T$ compared to the toy model, but crucially these differences do not seem to be related to the historical jet latitude.

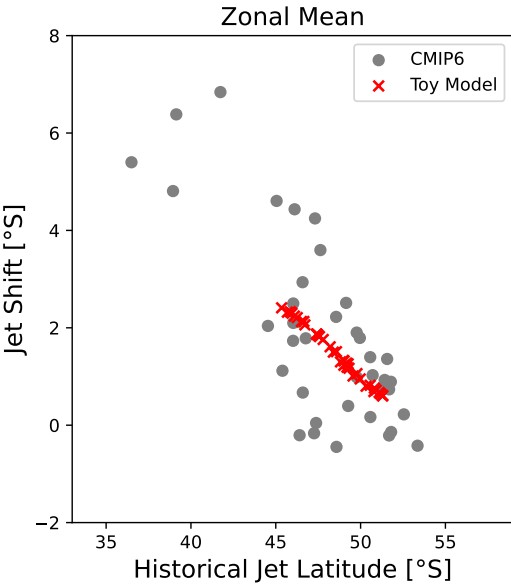

**Figure 7.** As in Fig. 2a with the results from the toy model added.

Furthermore, in the toy model the asymmetry between $\Delta U_P$ and $\Delta U_T$ in Fig. 6b has its origin in both halves of the hemisphere
(the plot would be qualitatively similar if only including the response of either half of the hemisphere). While CMIP6 models simulate a poleward shift of one of the jets in each basin (the equatorward one in the Pacific half and the single jet in the Indo-Atlantic half), we also observe a jet strengthening (the poleward one in the Pacific half and the single jet in the Indo-Atlantic half). This leads to the response peak ($\Delta U_P$) being stronger than the response trough ($\Delta U_T$).

Coming back to the EC, we reproduce Fig. 2a in Fig. 7 and also add the results from the toy model. We note that the toy model shows an EC. In the toy model, the EC arises from the anchored response, because the response is the same in all realisations, but it projects differently onto a shift depending on the zonal-mean climatological jet latitude. While this geometric argument was already proposed by Simpson and Polvani (2016), our toy model results suggest that the zonal asymmetry could ultimately be the origin of the anchored response, and thus of the EC.


We stress again that the full hemispheric zonal-mean jet latitude should not be interpreted as the latitude of a zonally coherent jet structure, but instead arises from averaging over different hemispheric regions that show different structures. This kind of measure and its associated EC should therefore be interpreted with caution. Ultimately, Fig. 2b demonstrates that in regions with only a single jet, there is no relationship between jet latitude and future jet shift, and furthermore we did not find
a relation when considering the jets of the double jet structure individually (not shown).

We note that Bracegirdle et al. (2013) (hereafter B13) analysed the relation between climatological jet position and future jet shift in the Atlantic, Indian and Pacific sectors separately in a CMIP5 ensemble. The authors acknowledged that the jet latitude might be difficult to define in the Pacific region, but used it as a measure nevertheless. They did find a correlation of $r = 0.39$ in the Atlantic sector in the annual mean, which was significant at the 5 % level. However, this is not necessarily in contradiction to our findings, given the use of annual averages. As in our present work, B13 found high correlation coefficients in the Pacific region both for individual seasons (except for DJF) as well as for the annual mean. Unfortunately no scatter plot was provided, except for the annual mean, making it hard to judge to which extent the seasonal results might be dominated by outliers, as in our findings (Fig. 2c). We further stress that, as acknowledged by B13, the jet position is difficult to define in the Pacific. The presence of an EC in the annual mean is consistent with the findings here, since it is an average over a double jet structure in winter and a single jet structure in summer. The mechanism would therefore be the same as for the zonal-mean results presented here.

## 4 Summary and Conclusion

We argue that the wintertime zonal-mean jet latitude should not be interpreted as a location measure of a zonally coherent structure, since it averages over a single jet structure in the Indo-Atlantic region and a double jet structure in the Pacific region. We demonstrate that a substantial amount of the historical jet latitude variation among the CMIP6 ensemble can be explained by relative differences in strength of the Pacific double jet structure. Using a simple toy model, we propose that the previously observed anchoring of the zonal-mean zonal wind response (i.e. the fact that the response structure does not track the climatological jet latitude; Simpson and Polvani, 2016; Simpson et al., 2021) is an artefact of the zonally asymmetric jet structure in CMIP5 and CMIP6 leading to large variations in historical jet latitude but comparatively little spread in response.

Following Simpson and Polvani (2016), we suggest that the previously identified EC between zonal-mean historical jet latitude and future jet shift can be explained in the context of the anchored zonal wind response projecting differently onto a jet shift, depending on the climatological jet latitude. However this implies that a larger zonal-mean jet shift does not signify a larger zonal wind response. The toy model demonstrates how the EC could arise from the zonal asymmetries without any direct causal link between zonal-mean jet latitude and future zonal wind changes. Regardless of the mechanism for the anchored wind response, the physical interpretation of the EC is unclear, since the measure of zonal-mean jet latitude is not reflecting a zonally coherent position. These conclusions are further supported by the fact that the EC only holds in the zonal mean and does not appear in the Pacific or Indo-Atlantic halves of the hemisphere separately.

Our results thus suggest that the apparent EC identified in previous work is caused by failing to properly account for the confounding factor of longitudinal asymmetry. This is a special case of a statistical phenomenon known as the Yule-Simpson effect (e.g. Goltz and Smith, 2010). Hence the results presented here demonstrate that caution is needed when using zonally

averaged metrics to interpret zonally asymmetric circulation features, such as the wintertime southern hemispheric jet. We therefore suggest moving away from the search for an EC on the zonal-mean circulation in austral winter, and instead focusing on individual hemispheric sectors with physically coherent circulation features. As shown here, the physical interpretation of the jet latitude is not always clear, quantifying the austral winter circulation response in terms of the strength of the zonal wind response, rather than as the change in jet latitude, might prove more informative.

**Appendix A: Models**

**Appendix B: Toy Model**

Here we show the same comparison of zonal-mean zonal winds from the CMIP6 ensemble with the toy model as in Fig. 5, but for the Indo-Atlantic sector (Fig. B1) and the full hemispheric zonal-mean (Fig. B2).

*Author contributions.* PB designed the toy model, performed the model analysis, and wrote the paper. Both PC and TGS contributed to the interpretation of the results and the writing of the paper.

*Competing interests.* At least one of the (co-)authors is a member of the editorial board of Weather and Climate Dynamics

*Acknowledgements.* We are grateful to Camille Li for a helpful discussion and also to Ed Gerber for very insightful comments. We thank two anonymous reviewers for their constructive criticism. Philipp Breul was supported by the Centre for Doctoral Training in Mathematics of Planet Earth and the Department of Mathematics at Imperial College London. This research was supported by the Engineering and Physical Sciences Research Council (grant no. EP/L016613/1) and the Natural Environment Research Council (grant no. NE/T006250/1). We acknowledge computational resources and support provided by the Imperial College Research Computing Service (http://doi.org/10.14469/hpc/2232).

| Model | Monthly | Daily | | | |
|---|:---:|:---:|---|:---:|:---:|
| AWI-CM-1-1-MR | x | | BCC-CSM2-MR | x | |
| CAMS-CSM1-0 | x | | CAS-ESM2-0 | x | |
| CESM2 | x | | CESM2-WACCM | x | x |
| CIESM | x | | CMCC-CM2-SR5 | x | x |
| CMCC-ESM2 | x | x | CNRM-CM6-1 | x | x |
| CNRM-CM6-1-HR | x | x | CNRM-ESM2-1 | x | x |
| CanESM5 | x | x | EC-Earth3 | x | x |
| EC-Earth3-CC | x | x | EC-Earth3-Veg | x | |
| EC-Earth3-Veg-LR | x | x | FGOALS-f3-L | x | x |
| FGOALS-g3 | x | x | FIO-ESM-2-0 | x | |
| GFDL-CM4 | x | x | GFDL-ESM4 | x | |
| HadGEM3-GC31-LL | x | x | HadGEM3-GC31-MM | x | x |
| IITM-ESM | x | x | INM-CM4-8 | x | x |
| INM-CM5-0 | x | x | IPSL-CM6A-LR | x | x |
| KACE-1-0-G | x | x | KIOST-ESM | x | |
| MIROC-ES2L | x | x | MIROC6 | x | x |
| MPI-ESM1-2-HR | x | x | MPI-ESM1-2-LR | x | x |
| NESM3 | x | x | NorESM2-LM | x | x |
| NorESM2-MM | x | x | TaiESM1 | x | x |
| UKESM1-0-LL | x | x | | | |

**Table A1.** The CMIP6 models that are used in this analysis; monthly data was available for all models but daily data for only a subset.

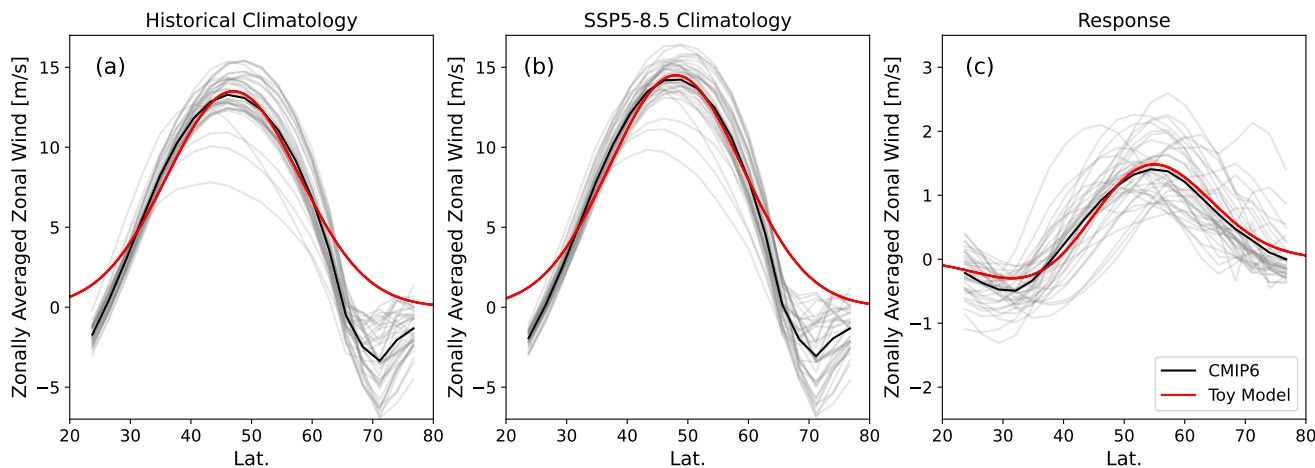

**Figure B1.** As in Fig. 5 but for the Indo-Atlantic sector (300°- 120°).

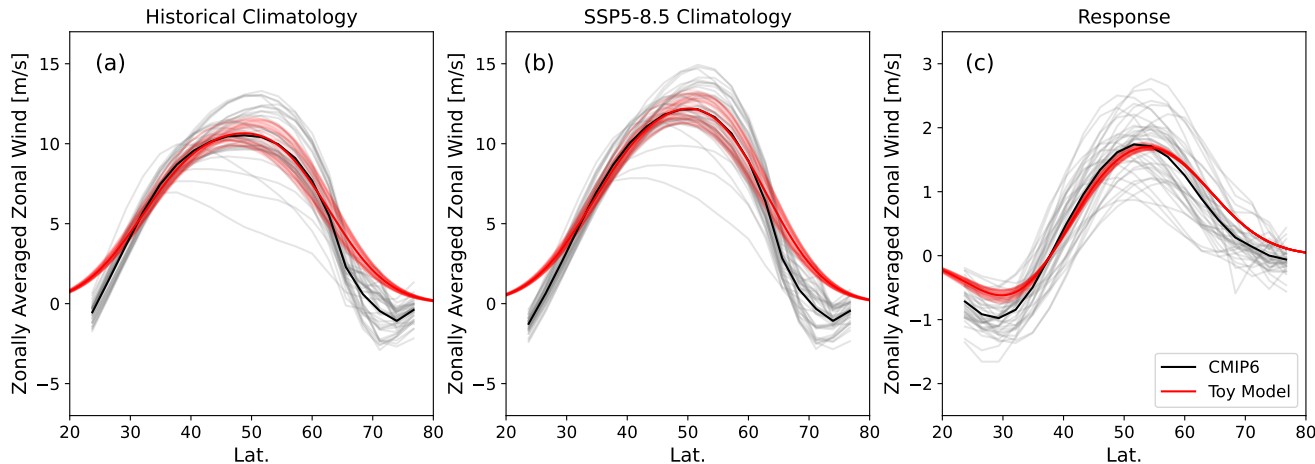

**Figure B2.** As in Fig. 5 but for the full-hemispheric zonal-mean.

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
