# Peer review of "Revisiting the wintertime emergent constraint of the Southern Hemispheric midlatitude jet response to global warming"

_Weather and Climate Dynamics, 2022_

## Author Comment (AC1)

**Final response to the reviewer comments on WCD-2022-42**

We thank both reviewers for their insightful and constructive comments. The reviewers' comments are marked in blue, while our responses are marked in black.

We want to briefly discuss the general changes we plan to make to the manuscript. On request of Reviewer 1, we expanded the analysis to include 39 CMIP6 models, and the conclusions remain valid. Further, we will add a figure showing the inter-model differences in zonal wind climatology regressed onto the inter-model differences in jet latitude to visualise the contributions from the different sectors. To further support the argument that the Pacific double jet structure influences the zonal-mean jet latitude, not only in the toy model but also in the CMIP6 ensemble, we will plot the difference in strength of the two Pacific jets versus the zonal-mean jet latitude. We find a strong connection between the two. Additionally, we will make changes to the text to communicate our line of reasoning more clearly.

**Reviewer 1**

Revisiting the wintertime emergent constraint of the Southern Hemispheric mid-latitude jet response to global warming, by Breul et al

This paper revisits the wintertime emergent constraint on the Southern Hemisphere (SH) jet latitude, which relates the climatological jet position to the future jet shift in CMIP models. Specifically, previous work has argued that this constraint arises primarily as a result of a geometrical effect related to the fact that the wind anomalies in the zonal mean are anchored at the same location regardless of the climatological jet position. It is argued here that the climatological jet latitude differences across models are related to inter-model differences in the relative strength of the single Atlantic jet and the strengths of the two jets in the double jet Pacific structure. The authors argue that the zonal mean jet location is not very physically meaningful given that it is an average over these two distinct jet structures. I think this study is tackling an issue that needed to be resolved and I find the arguments somewhat convincing. I do think that some improvement on the connections between their toy model and the actual CMIP model behavior could strengthen the conclusions considerably. At the moment, the toy model is presented as being able to reproduce the relationships that are found in CMIP, but it might be nice if there were a way to connect the toy model to the behavior of individual models a bit more. I've made some suggestions along these lines below, but overall I think this manuscript is acceptable for publication after minor revisions.

General suggestions:

(1) Improving the linkage between the toy model and the behavior of the individual CMIP models. At the moment, the pieces of evidence for the authors argument are

(a) there is no local connection between the jet position and jet latitude in the two longitudinal sectors separately and

(b) the toy model can exhibit similar behavior to the CMIP models when random values are added to the amplitude of a1 and a2. It's not totally clear to me what a1 and a2 represent but I'm assuming it's either the amplitude of the two pacific jets or the amplitude of one of the Pacific jets and the Atlantic jet (see comment below).

Anyway, the piece that seems a bit missing is then linking this back to the behavior of CMIP. It seems like it should be possible to then show that there is a relationship between the CMIP zonal mean jet latitude and the amplitude of the relevant jets in a manner that is similar to the toy model. If possible, I suggest the authors investigate whether these aspects can be tied together a bit better e.g., is the climatological latitude of the jet in each model highly correlated with the amplitude of the Pacific southern jet?

We thank the reviewer for their suggestions. We agree that the here proposed explanation suggests a connection between the strength of the Pacific jets and the zonal mean jet latitude. To be more specific, the difference in strength of the equatorward and poleward jets of the Pacific double jet structure influences the zonal mean jet latitude.

We analysed all models that showed two distinct peaks of zonal mean winds over the Pacific region (120° -- 300°) for the historical experiments, of which there were 16 in total. (Note however that this does not mean that the other models do not have a physical double jet structure. If one wishes to analyse more models, different sector definitions could potentially be helpful, otherwise a similar analysis with daily data could be done.) We found a strong correlation between the difference in jet strength of the Pacific double jets and the zonal mean jet position, which is in agreement with our proposed explanation.

[Figure]

To clarify point (a) of the reviewer, although this was not discussed in the original manuscript, we do find a connection between the two parts of the hemisphere. In Fig. R1 below we show a regression map of inter-model differences in zonal wind climatology versus the inter-model differences in zonal mean jet latitude; we plot the jet positions as black lines. We observe that the regression map is relatively zonally symmetric, with a slight poleward deviation in the Pacific region. However, the projection onto the jet climatology is very different in the two parts of the hemispheres. In the Indo-Atlantic the regression projects mostly onto a shift (with some strengthening) of the jet, while in the Pacific it projects more onto a strengthening of the poleward jet and a slight weakening of the equatorward jet, which is in line with the mechanism detailed above. A poleward shift of the Indo-Atlantic jet coupled with a strengthening/weakening of the poleward/equatorward Pacific jet therefore leads to the large spread in zonal mean jet latitude compared to the Indo-Atlantic half or the zonal mean in DJF. The overall argument stays the same, however. We will discuss this briefly in the paper.

[Figure]

Fig. R1. Inter-model differences in zonal wind climatology regressed onto the inter-model differences in zonal-mean jet latitude. The solid black lines show the jet positions.

(2) I think the prior work of Bracegirdle et al 2013 doi:10.1002/jgrd.50153 deserves some discussion. They showed that the emergent constraint holds in the Pacific sector and is kind of there, albeit weaker, in the Atlantic and the Indian ocean sector. The big difference here is probably that you are looking at the winter while Bracegirdle et al used the annual mean, but I think it could still be worth discussing this prior work and why your conclusions differ.

We thank the reviewer for making us aware of this relevant prior work. There are some differences in the methodology of the two studies; for example Bracegirdle et al 2013 (B13) use surface winds compared to 850 hPa winds, a CMIP5 ensemble compared to CMIP6, slightly differing sector definitions, and different averaging periods for historical and forced experiments. However, we do not expect these choices to change the results conceptually.

B13 find significant correlations between jet position and shift for the Pacific and Indian sectors for all seasons (except DJF) and the annual mean. We point out that we also find a high "significant" Pearson correlation coefficient for the Atlantic/Indian sector, but this is due to large outliers, due to the double jet structure making the definition of jet latitude problematic, as also acknowledged by B13. However, B13 only show scatter plots of jet position and jet latitude for the annual mean, but even here outliers are present in the Pacific sector. Averaging across seasons is problematic in a similar way to averaging across longitudes (as argued in this study), since it confounds a single jet structure that is mostly present in the summertime with the double jet structure in wintertime.

The weak but significant correlation in the Atlantic sector reported by B13 is different to our findings. However, as the reviewer pointed out, this is only present in the annual mean and SON. Given that the study calculates 20 correlation coefficients for the different seasons and sectors for jet position and jet shift, this is not necessarily in contrast to our findings.

Overall, we do not think the findings disagree. B13 acknowledged that the definition of jet latitude in the Pacific can be problematic but proceeded using it anyway. Apart from the Indian sector (which could be revisited in future work), the model analyses seem in agreement.

Minor comments by line number:

l20: It sounds a bit strange to first cite Simpson and Polvani 2016 and Breul et al 2022 in the context of studies that link the jet shift to annular mode timescale and then cite them in the next sentence to say that these studies couldn't find that constraint. Suggest removing Simpson and Polvani 2016 and Breul et al 2022 from the first lot of citations.

The paragraph will be rewritten.

l40: What has motivated this choice of 22 models? Why not use all the models?

We had easy access to these 22 models but did not expect qualitatively different findings since the results were in line with findings from Simpson et al. 2021. However, we have now expanded the study to include 39 models, and the prior findings are confirmed.

l46: "chose" --> "chosen"

Changed

l48: I think Kidston and Gerber (2010) used this quadratic method to define the jet latitude before Barnes and Polvani (2013) did. Suggest citing them instead.

That is right, will be changed.

l55: Simpson et al (2021) might be the more relevant one here since they used CMIP6, while Simpson and Polvani (2016) used CMIP5.

We agree that Simpson et al (2021) are more relevant, but Simpson and Polvani (2016) are the first to identify the constraint, so we will cite both.

l56: This correlation is also quite a lot higher than was found for CMIP6 by Simpson et al 2021 (they found -0.57). This is probably due to the different models being used, but some motivation for not including some models should probably be given here.

As mentioned above, we have now included 39 models but still find a higher correlation (r=-0.76). The difference could be due to Simpson et al. using 700 hPa zonal winds while we are using zonal winds at 850 hPa.

l90: I'm not sure that "observed response" is the best phrasing here. It could be mixed up with the climate change signal or the climatological jet latitude in the observations. I think really you're referring to the CMIP behavior here?

Agreed, we will change it to "explain the response shown by the CMIP ensemble".

l97 and 98: I may have missed it but I don't think you've defined a1 and a2. I'm assuming you're defining the amplitude of the three jets with a1, a2 and a3, but it's not clear to me which jets a1 and a2 correspond to.

We agree that this needs to be clarified.

l99: Something's not right about this sentence "and the therefore also" is not making sense.

Thank you, changed it to "and therefore also the response shown in"

**Reviewer 2**

This manuscript raises questions related to the physical interpretation of zonal mean jet in the wintertime Southern Hemisphere where the latitudes of jets in the Atlantic and Pacific differ from each other. The authors found from their CMIP6 multi-model analyses that the emergent constraint found for the zonal mean jet between its present-day and future shift does not hold for individual two halves of the hemisphere. Using a toy model, the authors demonstrate that a latitudinal shift of the zonal mean jet does not necessarily reflect a coherent jet shift in different longitudes, by showing that the strengthening/weakening of a zonally localized jet can also influence the zonal mean jet shift as a statistical artifact. Also, the toy model by the authors explains the similarity of the meridional structure of the future change of zonal mean zonal wind among CMIP6 models regardless of the present-day jet profile. I think this journal is appropriate for this article to be published after the following comments are addressed.

Major comments (Not necessarily in the order of importance)

a) Line 74.

The authors diagnose the physical existence of the jet using unfiltered daily data in Figures 2b-c. Instantaneous zonal wind in the southern hemisphere mid-latitudes is sensitive to the synoptic scale cyclones/anticyclones moving eastward due to the background jet stream (Figure 2a). Therefore, the physical meaning of zonal wind between Figure 2a and Figure 2b-c is different; the former is the basic state component, and the latter is the mixture of basic state and eddy components. For the author's purpose, they should use daily data after applying a low-pass filter of e.g. 8 days to filter out the eddy component.

We thank the reviewer for their suggestions. We agree with the reviewer and made the suggested change. We did not find substantial changes.

b) Figure 4c, Figure 5

These figures are based on the toy model where random variables are added to a1 and a2 respectively. Please explain why the values added to a1 and a2 are set independently of each other (i.e. no correlation) in comparison to the CMIP6 results

Below we plot the climatological jet strengths of the Pacific double jet structure against each other for those models that show two peaks in the time and zonal mean over the extended Pacific region (120° -- 300°), i.e. the CMIP6 models with a double peak in Fig. 4a. We do not observe any clear relationship between the two jet strengths. Ultimately we chose to set values for a1 and a2 randomly to reproduce the spread observed in CMIP6, while at the same time making as few assumptions as possible.

[Figure]

c) Figure 5a

Φp of CMIP6 models is less clearly anchored, as the authors also discuss in line 135. There rather seems like two groups of CMIP6 models; one group is non-anchored models and the other is anchored models following the identity line in Figure 5a. The authors should explain what caused the second group. Given the fact that such a case never appeared in the toy model (Figure 5a), I wonder to what extent the toy model is applicable to interpret the anchoring found for CMIP6 models.

Using the expanded dataset, we do not see a division into two groups. However, we agree that there could be some remaining weak connection of Φp to the zonal mean jet latitude, and the Pearson correlation coefficient is r=0.4 (p=0.02). We believe that this could come from the response in the Indo-Atlantic region having some connection to the Indo-Atlantic jet latitude, which could translate into a weak remaining connection to the zonal mean. At the same time, the toy model does show a weak connection between Φp and the zonal mean jet latitude even without accounting for this effect.

We decided to cut this paragraph since we believe it distracts from the main story.

d) line 150

While Figures (3) and (A2) conceptually explain that the inter-model difference of the zonal mean jet axis is caused by the Pacific double jet structure, there is no figure showing it is the case in CMIP6. To justify the conceptual argument, please add a panel in Figure 3 which is like the current Figure 3 but for actual CMIP6 models.

We have plotted the difference in strengths of the Pacific double jets against the zonal mean jet latitude and find a strong connection between the two. Please see the first answer to Reviewer1's comments, which addresses this concern in detail.

Minor comments

a) Equation (1)

Please refine it in a form including subscripts (1,2) to represent the Pacific jet described in Table (1).

Every jet is represented by a structure described by eq. 1; using subscripts in this equation is therefore not appropriate. We have clarified this in the text.